# Three Ecological Models to Evaluate the Effectiveness of *Trichoderma* spp. for Suppressing Aflatoxigenic *Aspergillus flavus* and *Aspergillus parasiticus*

**DOI:** 10.3390/toxins16070314

**Published:** 2024-07-12

**Authors:** Nataliia Voloshchuk, Zilfa Irakoze, Seogchan Kang, Joshua J. Kellogg, Josephine Wee

**Affiliations:** 1Department of Food Science, The Pennsylvania State University, University Park, PA 16802, USA; nvv5125@psu.edu (N.V.); zirakoze2@psu.edu (Z.I.); 2Department of Plant Pathology and Environmental Microbiology, The Pennsylvania State University, University Park, PA 16802, USA; sxk55@psu.edu; 3One Health Microbiome Center, HUCK Institutes of the Life Sciences, The Pennsylvania State University, University Park, PA 16802, USA; jjk6146@psu.edu; 4Department of Veterinary and Biomedical Sciences, The Pennsylvania State University, University Park, PA 16802, USA

**Keywords:** aflatoxin, *Trichoderma*, biocontrol, interaction, volatile metabolites, non-volatile metabolites

## Abstract

Chemical pesticides help reduce crop loss during production and storage. However, the carbon footprints and ecological costs associated with this strategy are unsustainable. Here, we used three in vitro models to characterize how different *Trichoderma* species interact with two aflatoxin producers, *Aspergillus flavus* and *Aspergillus parasiticus*, to help develop a climate-resilient biological control strategy against aflatoxigenic *Aspergillus* species. The growth rate of *Trichoderma* species is a critical factor in suppressing aflatoxigenic strains via physical interactions. The dual plate assay suggests that *Trichoderma* mainly suppresses *A. flavus* via antibiosis, whereas the suppression of *A. parasiticus* occurs through mycoparasitism. Volatile organic compounds (VOCs) produced by *Trichoderma* inhibited the growth of *A. parasiticus* (34.6 ± 3.3%) and *A. flavus* (20.9 ± 1.6%). The VOCs released by *T. asperellum* BTU and *T. harzianum* OSK-34 were most effective in suppressing *A. flavus* growth. Metabolites secreted by *T. asperellum* OSK-38, *T. asperellum* BTU, *T. virens* OSK-13, and *T. virens* OSK-36 reduced the growth of both aflatoxigenic species. Overall, *T. asperellum* BTU was the most effective at suppressing the growth and aflatoxin B1 production of both species across all models. This work will guide efforts to screen for effective biological control agents to mitigate aflatoxin accumulation.

## 1. Introduction

Aflatoxins (AFs) are a group of secondary metabolites predominantly produced by toxigenic *Aspergillus flavus* and *A. parasiticus* that contaminate crops during growth, harvest, and storage [1,2,3]. One of the competitive advantages is their ability to withstand wide fluctuations in temperature and humidity [4,5]. Their distribution varies according to crop and geographical location. *Aspergillus flavus* predominantly produces AFB1 in soil and on plants, whereas *A. parasiticus* can produce AFB1, AFB2, AFG1, and AFG2, and is more widespread in soil [6,7]. *A. flavus* appears to be more prevalent in agricultural soils used for food and feed crops compared to *A. parasiticus* [8]. Although *A. flavus* and *A. parasiticus* are ubiquitous in soil and plant environments, fungal growth and toxin production largely depend on the environmental conditions. Extreme weather conditions, such as high temperatures and drought, can cause plant stress, which results in an increased vulnerability to plant infection, with subsequent fungal infection and AF accumulation [1,4,9,10,11]. These climate scenarios could result in increasing AF hotspots, with predictions indicating that AF contamination will intensify over the next century [12,13]. In addition, the genetic diversity and geographical distribution of *A. flavus* populations also affect the distribution and severity of AF contamination [14]. Thus, approaches to AF mitigation should consider the physiology, distribution, and virulence of both *A. flavus* and *A. parasiticus*.

Many efforts have focused on developing strategies to mitigate AF contamination during the pre- and post-harvest stages. The reduction in AF contamination can be achieved by the displacement of AF-producing strains due to competition in the field. The most successful competitive exclusion strategy is the introduction of endemic non-aflatoxigenic genotypes of *A. flavus* that can outcompete the toxigenic strains. The effectiveness of non-aflatoxigenic *A. flavus* depends on geographical location. Non-aflatoxigenic *A. flavus* can displace populations of toxigenic strains of target crops in the field and reduce AF levels through robust propagation and rapid colonization [15]. For example, AF36 (NRRL 18543) has been tested on cotton, and AF36 Prevail^®^ has been tested on maize, pistachio, almonds, and figs [16]. Soil treatment, with a combined inoculum of non-aflatoxigenic strains of *A. flavus* (NRRL 21882) and *A. parasiticus* (NRRL 21369), significantly reduced the pre-harvest AF contamination of peanuts by 74.3–99.9% [17,18,19]. Although the use of non-aflatoxigenic strains to competitively exclude toxigenic strains has been effective, alternative biological control approaches to reduce AF contamination could strengthen and diversify the current mitigation strategies.

*Trichoderma* species have a long history of effective biological control for plant protection, growth promotion, and the suppression of plant pathogens [20,21,22,23,24]. *Trichoderma* species are fast-growing, ubiquitous, and abundant in soil and plant environments across diverse ecosystems, including agricultural soils. *Trichoderma* species are widely used as bio-fungicides, notably due to the production of volatile or non-volatile organic compounds with antifungal and antibiotic properties [24]. Some examples of volatile organic compounds (VOCs) produced by *Trichoderma* include alcohols, terpenes, sesquiterpenes, and aldehydes, while the secreted non-volatile metabolites include polyketides and peptaibols [25]. These bioactive compounds can function as chemical signals that enhance plant resistance to biotic and abiotic factors. Thus, *Trichoderma* species are popular biological control agents (BCAs) due to their physiological and genetic versatility, and metabolic diversity [26,27,28,29,30,31].

Abiotic stresses, caused by increasing global surface temperatures, record droughts, and intense rainfall, adversely affect soil and plant root ecosystems. To date, *Trichoderma*-based BCAs and biologicals are some of the most common bioproducts used in agriculture, promising climate-smart alternatives to sustain crop productivity, soil health, and plant pathogen control [27]. For example, biopriming wheat seeds with *T. harzianum* significantly enhanced plant drought tolerance compared to untreated seeds, reducing leaf wilting and increasing root vigor [32]. Additionally, *Trichoderma* treatment of *Arabidopsis thaliana* leaf tissue was able to mitigate drought, heat, and a drought/heat combination by sustaining plant biomass and yield [33]. Certain *Trichoderma* isolates can grow and survive at wide temperature ranges between 25 and 40 °C, suggesting their diverse ecological niches [34]. Both *Trichoderma* and aflatoxigenic *Aspergillus* species co-exist in soil and plant root ecosystems, thus, understanding their interactions could facilitate the development of biocontrol strategies that simultaneously enhance crop resistance to abiotic stresses and diseases, including those caused by aflatoxigenic *Aspergillus* species. Such strategies could contribute to ensuring food security and safety, which is one of the challenges posed by climate change.

Previous studies have demonstrated the potential of *Trichoderma* spp. as BCAs to suppress the growth of aflatoxigenic *Aspergillus* and AF biosynthesis. In vitro studies using *T. harzianum* and *T. viride* suggested the production of extracellular enzymes, antibiotics, and VOCs that suppress *A. flavus* growth [35,36]. Ren et al. (2022) showed that 20 different strains of *Trichoderma* inhibited the growth of *A. flavus* and AF accumulation at varying degrees, and this inhibition is independent of AF gene expression [37]. Boukaew et al. (2023) demonstrate that volatile and non-volatile metabolites of *T. asperelloides* play an important role in the control of aflatoxigenic *Aspergillus* growth and AF production in vitro and on peanuts [38]. The pre-harvest treatment of peanuts and sweet corn with *Trichoderma*-based BCAs inhibited *A. flavus* plant infection and reduced AF contamination [39,40,41]. *Trichoderma* strains were found to effectively inhibit *A. flavus* infection in the field and during storage, reducing mold density and AFB1 production to a level below the public health concern [42]. These studies indicate that *Trichoderma* could potentially interact with *Aspergillus* in the following ways: (1) through competitive exclusion as a result of limited resources, (2) growth inhibition via antibiosis (production of volatile or non-volatile metabolites by one fungus with the ability to suppress another), and (3) mycoparasitism (the ability of one fungus to parasitize the other). Therefore, there is an interest in using *Trichoderma* as a BCA for the suppression of aflatoxigenic fungal growth and AF contamination. However, the mechanism of *Trichoderma*–*Aspergillus* interactions, even under in vitro laboratory conditions, has not been well characterized.

The aim of this study was to characterize *Trichoderma*–*Aspergillus* interactions through physical contact, volatile-mediated interactions, and the secretion of metabolites. We hypothesized that the *Trichoderma* strains with the highest growth rates would be the most successful through these modes of interaction. We further hypothesized that there would be strain-specific differences in the effects of *Trichoderma* VOCs and non-VOCs on *Aspergillus* growth and toxin production. We tested the three ecological models to evaluate the effectiveness of diverse *Trichoderma* spp. for suppressing aflatoxigenic *A. flavus* and *A. parasiticus*. Fungal growth and aflatoxin levels were used to evaluate the effectiveness of the dual plate (physical interaction), sandwiched plate (VOC-mediated interaction), and cellophane membrane-based (interaction via secreted metabolites) assays.

## 2. Results

### 2.1. Growth Rate as a Pre-Requisite of Strain Fitness during Interaction Studies

The ability to grow rapidly is one essential characteristic of effective *Trichoderma* strains used as current BCAs [43]. We hypothesize that *Trichoderma* with the highest growth rates will be the most effective in regulating fungal growth and toxin production in all three ecological models. Growth rate was measured as a function of colony diameter over 24, 48, and 72 h on solid PDA medium at 30 °C (Figure 1). Six out of eight tested *Trichoderma* exhibited rapid growth, reaching ~67–84 mm in colony diameter by 72 h. Fast-growing strains included *T. asperellum* OSK-38, *T. asperellum* BTU, *T. harzianum* OSK-21, *T. harzianum* OSK-34, *T. virens* OSK-13, and *T. virens* OSK-36. In contrast, *T. atroviride* PNB 12-IAI exhibited similar growth rates when compared with aflatoxigenic *A. flavus* (A.f. 3357) and *A. parasiticus* (A.p. B62). Finally, *T. viride* OSK-22 demonstrated the slowest growth when compared to all other strains tested. Taken together, after 24 h, all fast-growing *Trichoderma* strains produced an average colony diameter of 20 ± 2 mm, which increased 3-fold by 48 h to 59 ± 2 mm. By 72 h, colony diameter increased to an average of 80 ± 3 mm.

### 2.2. Dual Culture Interaction Assay between Trichoderma and Aspergillus

The physical interactions between *Trichoderma* with A.f. 3357 and A.p. B62 were assessed using a dual culture plate assay, where both fungi were grown in the same physical space. Six out of eight fast-growing *Trichoderma* reduced the colony radius of A.f. 3357 and A.p. B62 when compared to the control at 120 h (Figure 2A,C). In contrast, the colony radius of aflatoxigenic strains in the presence of moderate growing *T. atroviride* PNB 12-IAI and slow-growing *T. viride* OSK-22 were not significantly different than the control. In support of colony radius reduction, a 38–42% growth inhibition was observed when A.p. B62 was grown in the presence of six *Trichoderma* spp. while a 40–45% growth inhibition was observed in the interaction with A.f. 3357 (Figure 2B,D and Table 1). The highest percentage growth inhibition was observed between *T. harzianum* OSK 34 with A.p. B62 (42.4 ± 2.2%) and between *T. virens* OSK-13 with A.f. 3357 (45.1 ± 0.9%). In contrast, moderate to slow-growing *T. atroviride* PNB 12-IAI and *T. viride* OSK-22 exhibited minimal growth inhibition when tested with both aflatoxigenic strains (0.6–5.5%). The ability of *T. asperellum* OSK-38, *T. asperellum* BTU, *T. harzianum* OSK-21, *T. harzianum* OSK-34, *T. virens* OSK-13, and *T. virens* OSK-36 to inhibit the growth of A.p. B62 was similar and consistent with the magnitude of growth inhibition of A.f. 3357 (Figure 2B,D).

In addition to percent growth inhibition, the type of antagonistic interaction between *Trichoderma* and *Aspergillus* was assessed as described by Whipps et al. (1987) (Table 1) [44]. It appears that *Trichoderma* predominantly drives type 1–2 interactions with A.p. B62, while types 3–4 are more common between *Trichoderma* and A.f. 3357. For example, six *Trichoderma* species mutually developed with A.p. B62, characteristic of type 1–2 interactions where *Trichoderma* mycelium was observed to overgrow the *Aspergillus* colony (Table 1, Figure 2E columns 1 and 2). In contrast, the formation of a zone of inhibition was observed between A.f. 3357 and *Trichoderma*, characteristic of type 3–4 interactions (Table 1, Figure 2E column 3). Slow-growing *T. viride* OSK-22 had no significant impact on the growth of *Aspergillus* compared to other fast-growing *Trichoderma*. Of note, the dual culture plate assay did not visibly reduce norsolorinic acid accumulation at the bottom of A.p. B62 colonies, as evidenced by the production of the bright orange pigment across all representative images (Figure 2E column 2).

### 2.3. Sandwiched Plate Interaction Assay to Assess the Effect of Volatile Organic Compounds between Trichoderma and Aspergillus

A sandwiched plate assay was used to assess the ability of VOCs produced by *Trichoderma* to suppress the growth of A.p. B62 and A.f. 3357. Based on colony diameter, all *Trichoderma* analyzed emitted VOCs with an inhibitory effect on A.p. B62 fungal growth when compared to the control (Figure 3A). In contrast, all strains except for *T. atroviride* PNB 12-IAI and *T. viride* OSK 22 significantly reduced the colony diameter of A.f. 3357 compared to the control (Figure 3D). For *Trichoderma* interactions with A.p. B62, the highest growth inhibition was observed by VOC produced by *T. harzianum* OSK-34 (44.0 ± 2.5%), followed by *T. virens* OSK-13 (34.7 ± 2.7%), *T. harzianum* OSK-21 (34.6 ± 3.4%), *T. asperellum* BTU (33.9 ± 3.1%), and *T. virens* OSK-36 (32.5 ± 3.0%) (Figure 3B). In contrast, when assessing the interaction with A.f. 3357, VOCs produced by *T. asperellum* BTU had the highest growth inhibition (28.0 ± 1.9%), followed by both *T. harzianum* OSK-34 (23.1 ± 1.2%) and OSK-21 (20.1 ± 1.5%) (Figure 3E). Of note, A.p. B62 demonstrated higher sensitivity to VOC produced by *Trichoderma* (growth inhibition range, 28.3–44.0%) when compared to A.f. 3357 (16.8–28.0%). The VOCs produced by slow-growing strains of *T. atroviride* PNB 12-IAI and *T. viride* OSK-22 had minimal effects on the growth inhibition of A.p. B62 and A.f. 3357.

To examine whether *Trichoderma* VOCs can regulate AF production in *Aspergillus*, AF secreted into the medium with and without the presence of *Trichoderma* VOCs was assessed by TLC (Figure 3C,F). *Aspergillus flavus* predominantly produces AFB1 (the most potent AF), while *A. parasiticus* produces AF B1, B2, G1, and G2. It is important to note that since most *Trichoderma* spp. reduced the colony diameter of *Aspergillus*, a reduction in toxin levels as observed by TLC is dependent on fungal growth. Thus, TLC was used to validate toxin levels to support the growth reduction observed. The *Trichoderma* VOC-mediated changes on AFB1 levels were strain and species-specific. When compared to the control in the case of A.p. B62, the VOCs emitted by both *T. harzianum* OSK-21 and OSK-34 showed no observable AFB1, AFB2, and AFG1 bands. We observed a similar trend with the VOCs produced by *T. asperellum* BTU, while the toxin levels of other *Trichoderma* strains were similar to the control and standard. Observed levels of AFB1 with TLC corresponded to the intensity of orange pigment present in the fungal colony of A.p. B62 (Figure 3G). A.f. 3357 was more resistant to the VOC produced by *Trichoderma*, as evidenced by growth inhibition. The highest % inhibition was observed between *T. asperellum* BTU and A.f. 3357 (28 ± 1.9%), which is supported by the observed reduction in AFB1. In addition, A.f. 3357 with *T. virens* OSK-13, *T. asperellum* OSK-38, *T. harzianum* OSK-21, and *T. viride* OSK-22 also demonstrate reduced AFB1 levels when compared to the control (Figure 3E–G).

### 2.4. Cellophane Membrane-Based Interaction Assay between Trichoderma and Aspergillus

The effect of the non-volatile metabolites or secreted metabolites of *Trichoderma* on the fungal growth and toxin production of *Aspergillus* was assessed using a cellophane membrane-based assay. Seven out of eight *Trichoderma* strains secreted metabolites into the growth medium that demonstrated a statistically significant reduction in the colony diameter of A.p. B62 compared to the control (Figure 4A). In contrast, the secreted metabolites of 5 out of 8 *Trichoderma* strains reduced the colony diameter of A.f. 3357 (Figure 4D). The growth of A.p. B62 was most inhibited by the secreted metabolites of *T. virens* OSK-36 (88.4 ± 3.6%) and *T. asperellum* BTU (87.8 ± 1.9%), followed by *T. virens* OSK-13 (71.1 ± 7.8%) and *T. asperellum* OSK-38 (70.4 ± 2.9%) (Figure 4B). In contrast, A.f. 3357 demonstrated the highest sensitivity to the secreted metabolites produced by *T. asperellum* BTU (100.0 ± 0%; no detectable growth of A.f. 3357) and *T. asperellum* OSK-38 (89.5 ± 4.9% inhibition) (Figure 4E). It appears that A.f. 3357 was more resistant to the metabolites secreted by *T. virens* OSK-36 (64.9 ± 1.5%) and *T. virens* OSK-13 (62.5 ± 4.6%) compared to A.p. B62. The secreted metabolites produced by *T. viride* OSK-22 had minimal effect on the growth inhibition of A.p. B62 and A.f. 3357. In sum, this model that assesses the effect of secreted metabolites produced by *Trichoderma* was most effective, compared to the dual and sandwiched plate assays, in inhibiting the growth of A.p. B62 (70.4–88.4% inhibition) and A.f. 3357 (62.5–100.0% inhibition).

To assess the effect of the secreted metabolites of *Trichoderma* on the AF levels produced by *Aspergillus*, AFs were extracted from the media on day 5 and analyzed using TLC (Figure 4C). In assessing the interaction between A.p. B62 and *Trichoderma*, the intensity of AF bands supports the percent growth inhibition by secreted metabolites. Specifically, the highest percent inhibition was observed with *T. virens* OSK-36, *T. asperellum* BTU, *T. virens* OSK-13, and *T. asperellum* OSK-38. This corresponded to the absence or reduced intensity of AF bands when compared to the control (Figure 4C). On the other hand, *T. viride* OSK-22 did not significantly alter the colony diameter of A.p. B62 (4.2 ± 0.9% inhibition), consistent with the presence of AF bands. With A.f. 3357-*Trichoderma* interactions, the absence of an AFB1 band by *T. asperellum* OSK-38 and *T. asperellum* BTU is dependent on growth suppression. *T. virens* OSK-13 and *T. virens* OSK-36 inhibited the growth of A.f. 3357 (60%), consistent with non-detectable AFB1 bands (Figure 4F). Finally, despite minimal growth inhibition, in *T. atroviride* PNB 12-IAI, *T. harzianum* OSK-21, and *T. harzianum* OSK-34, a reduction in the intensity of AFB1 bands was observed. The band intensity of OSK-22 was not visibly different compared to control. The observed levels of AFB1 correspond with the colony diameter and intensity of the orange-colored norsolorinic acid at the bottom of the A.p. B62 colonies (Figure 4G column 2). It appears that the influence of *Trichoderma* non-volatile metabolites on the production of AFB1 was strain-specific and consistent between A.p. B62 and A.f. 3357 (Figure 4C,F).

## 3. Discussion

Global climate change, characterized by extreme weather conditions, can accelerate plant stress, resulting in an increased vulnerability to fungal infection and AF accumulation. *Trichoderma* species are widely distributed in soil and plant environments and are physiologically, genetically, and metabolically diverse [24]. *Trichoderma* species evolved to adapt to different ecosystems, including those inhabited by aflatoxigenic *Aspergillus*, and thus developed chemical ecology-based strategies for fitness and survival. Chemical signals can enhance the resistance to abiotic and biotic factors, including plant pathogens. This could help explain why approximately 60% of the registered products for plant promotion use *Trichoderma*. An excellent review by Dutta et al. (2022) discussed laboratory research for industrial interest on the physiology, mode of action, and application of *Trichoderma* for plant protection [24]. Taken together, the ecology and abundant knowledge of *Trichoderma* species make these fungi excellent candidates for the biocontrol of *Aspergillus* growth and the reduction in AF production.

Here, we combined three in vitro models to characterize the type of interaction between selected *Trichoderma* species and aflatoxigenic *A. flavus* and *A. parasiticus*. Using the dual culture plate assay, antagonism was assessed by the percentage of growth inhibition and type of colony interaction. Interestingly, *Trichoderma* spp. likely interact with *A. flavus* via antibiosis, whereas their interaction with *A. parasiticus* occurs via mycoparasitism. This interpretation is based on the formation of an inhibition zone classified as type 3 or 4 behavior (antibiosis) between *Trichoderma* and *A. flavus*, while *Trichoderma* mycelium overgrows the mycelium of *A. parasiticus* (classified as type 1 or 2 behavior; mycoparasitism). In support of our observations, Ren et al. (2022) observed predominantly type 3 or 4 interactions between *Trichoderma* spp. and *A. flavus* Af-9 [37]. In contrast, Boukaew et al. (2023) showed predominantly mycoparasitic activity between *T. asperelloides* on *A. flavus* PSRDC-4 and *A. parasiticus* TISTR 3276 [38]. One possible explanation for this observation is the levels of toxins and other secondary metabolites produced by different *A. parasiticus* strains compared to *A. flavus* strains. Since B62 is a genetic mutant that accumulates the orange pigment norsolorinic acid, AF levels in B62 are lower than in A.f. 3357. Thus, the presence of a zone of inhibition could indicate that *Trichoderma* spp. produce metabolites in defense against the increased AF production by A.f. 3357 compared to A.p. B62.

An assessment of the interaction between *Trichoderma* VOCs and *Aspergillus* growth demonstrated moderate growth inhibition (20–40% inhibition) using the sandwiched plate assay. This degree of moderate inhibition (~20–40%) by VOCs was also observed between *T. harzianum* and *T. viride* on five plant pathogens, *Alternaria alternata*, *A. solani*, *A. brassicae*, *Fusarium oxysporum*, and *F. solani* [22]. In another study, the VOCs produced by *T. asperelloides* significantly suppressed the mycelial growth of *A. parasiticus* and *A. flavus* by 75.67% and 71.33%, respectively [42]. *Trichoderma* species can produce VOCs that serve as mediators of signaling in the gas phase. However, the precision of volatile-mediated signaling depends on specific fungal–fungal interactions. In this study, we also observed differences between growth inhibition and AF production of A.f. 3357 by *T. asperellum* BTU and *T. harzianum* OSK-34. Both *Trichoderma* strains inhibited the growth of A.f. 3357 by ~30%. However, we observed higher levels of AFB1 with *T. harzianum* compared to *T. asperellum*. These results support the findings of Ren et al. (2022) that *Trichoderma* strains can suppress the growth of *A. flavus*, reduce toxin production, or both [37]. Future studies that compare the VOC profiles between *T. harzianum* and *T. asperellum* in addition to assessing AF gene expression and protein levels may help explain the differences observed in AFB1 levels.

In addition to the ability of *Trichoderma* to grow rapidly, these soilborne fungi are superior competitors due to the diversity and abundance of secreted metabolites. The secreted non-volatile metabolites of *Trichoderma* were effective in inhibiting the growth of both aflatoxigenic fungi. However, similar to the dual culture and sandwiched assay, the magnitude of growth inhibition was *Trichoderma* strain-specific, and dependent on *Trichoderma* strain fitness. It is interesting to note that the secreted metabolites produced by both *T. asperellum* strains, BTU and OSK-38, resulted in the significant growth inhibition of A.f. 3357 (89.5–100.0% inhibition). To our knowledge, no studies have characterized the interaction between *T. asperellum* and aflatoxigenic *Aspergillus* spp. Based on preliminary studies, the secreted metabolites of *Trichoderma* into PDA growth medium were more effective at inhibiting the growth of *Aspergillus* at 48 h compared to 30 h. As in other studies, it is important to note that the timing of secreted metabolite production is influenced by the growth substrate. PDA is a complex and rich growth substrate compared to the soil environment. Thus, secreted metabolite production on PDA may not be reflective of the soil environment, where nutrients are often limiting. In subsequent studies, we will compare the effects of different types and amounts of individual nutrients on the interaction between *Trichoderma* and *Aspergillus*.

Despite the combination of in vitro models that represent three different modes of action (physical interaction, VOC-mediated, and secreted metabolites), our models have some limitations. Our screening of *Trichoderma* did not account for other abiotic and biotic parameters that could be present in soil or plant environments. For example, changes in soil nutrient composition, temperature, or water availability. Similar to other studies, the ecological relevance of these models and their application in solving field problems requires further study. Nevertheless, these models provide a first step in screening hundreds or thousands of *Trichoderma* isolates from environmental samples for the initial characterization and selection of the most promising *Trichoderma* strains prior to field studies. In addition, the AF levels determined by semiquantitative TLC should be validated using HPLC. Finally, to understand specific *Trichoderma*–*Aspergillus* interactions, VOC profiling can be conducted using HS-SPME-GC/MS, and secreted metabolites can be profiled using LC-MS/MS. Evaluating the precise metabolomic-driven interaction between *Trichoderma* and *Aspergillus* would be the next necessary step to advance our understanding of the mechanisms that drive the successful biocontrol of aflatoxigenic *Aspergillus* using *Trichoderma*.

## 4. Conclusions

Consistent with previous studies, a high growth rate was characteristic of *Trichoderma*, and increased strain fitness drove successful interactions with *A. flavus* and *A. parasiticus*.

With the fast-growing trait as a pre-requisite for strain fitness and selection, a comparison across all three models (dual plate, sandwich, and cellophane) ranked *T. asperellum* BTU, *T. asperellum* OSK-38, and *T. virens* OSK-13 as the most successful in the interaction with *A. flavus* and *A. parasiticus*. These strains of *Trichoderma* were similar based on growth rate, % growth inhibition in the dual assay, and the ability of their non-volatile metabolites to regulate toxin production. However, the *Trichoderma*–*Aspergillus* interaction differs based on the type of interaction (antibiosis vs. mycoparasitism) and VOC-mediated signaling. When comparing all models between *A. flavus* and *A. parasiticus*, the effect of the secreted metabolites produced by *T. asperellum* BTU was the most effective and consistent relative to growth inhibition and AFB1 production.

The simultaneous use of these three models provides a framework for understanding the mode of action of *Trichoderma* spp. in regulating the growth and toxin production of aflatoxigenic *Aspergillus* spp. In vitro laboratory models that compare fungal–fungal interactions, as well as the effects of VOCs and non-VOCs, could help with the screening of a large library of *Trichoderma* strains relatively fast and with lower costs compared to field studies. The outcomes of this study could provide insights for developing biocontrol approaches to reduce and control AF production beyond current control mitigation strategies.

## 5. Materials and Methods

### 5.1. Fungal Cultures and Growth Conditions

Laboratory assays were conducted with 8 different *Trichoderma* strains: *T. asperellum* OSK-38, *T. asperellum* BTU, *T. atroviride* PNB12-IAI, *T. harzianum* OSK-21, *T. harzianum* OSK-34, *T. virens* OSK-13, *T. virens* OSK-36 and *T. viride* OSK-22. Two aflatoxigenic *Aspergillus* strains were used for interaction studies: *Aspergillus parasiticus* B62 (A.p. B62) and *A. flavus* NRRL 3357 (A.f. 3357). A.p. B62 was derived from the parent strain SU-1 (ATCC 56775), a wild-type AF producer. B62 accumulates norsolorinic acid, an orange-colored precursor of AF that helps in the rapid visualization of AF production [45,46]. A.f. 3357 is an AF-producing strain isolated from peanuts and obtained from the Agricultural Research Service (ARS) Culture Collection (Peoria, IL, USA). Strain designation, isolation information, and source are provided in Appendix A. All *Trichoderma* and *Aspergillus* strains used in this study were validated by Sanger sequencing using the internal transcribed spacer (ITS) region and the primers ITS1F/ITS4R (ITS1F: 5′-TCCGTAGGTGAACCTGCGG-3′and ITS4: 5′-TCCTCCGCTTATTGATATGC-3′). When species identification was not successful with ITS PCR, strains were confirmed with Tef1 primers for *T. asperellum* (Tef1F: 5′-CTCTGCCGTTGACTGTGAACG-3′ and Tef1R: 5′-CGATAGTGGGGTTGCCGTCAA-3′), *T. harzianum* (Tef1F: 5′-CCTCGATTCTCCCTCCACAT-3′ and Tef1R: 5′-GGCAATGATGAGGATAGCG-3′), *T. virens* (Tef1F: 5′-CCGTTTGATGCGGGGAGTCTA-3′ and Tef1R: 5′-GGCAAAGAGCAGCGAGGTA-3′), and *T. viride* (Tef1F: 5′-CAACTTTTCCCTCGCAGCAT-3′ and Tef1R: 5′-GAGGGTCGT TCTTGGAGTCA-3′) [47]. All fungal strains were grown on potato dextrose agar (PDA) plates (Difco™, Becton, Dickinson and Company, Sparks, MD, USA) at 30 °C for 5 days and maintained as spore stocks in 25% glycerol at −80 °C.

### 5.2. Dual Culture Plate Interaction Assay

Direct physical interaction (also known as the confrontation test) between *Trichoderma* and *Aspergillus* was studied on 85 mm Petri plates containing 20 mL of PDA. Five μL of spore suspension containing 1 × 10^6^ spores/mL of *Trichoderma* spp. were point inoculated 1 cm from the edge of the plate on one side and at the opposite side with the same number of spores of *Aspergillus* spp. (Figure 5A). The final spore concentration was 5 × 10^3^ spores. For control plates, either *Trichoderma* or *Aspergillus* was point-inoculated 1 cm from the edge of the plate. All co-culture and control plates were incubated at 30 °C for 5 days. The radial growth of *Trichoderma* and *Aspergillus* on co-culture and control plates was measured daily. Three plates for each co-culture and control were used in three independent biological experiments (total, *n* = 9).

The percent inhibition (PI) of radial growth observed in co-cultures of A.p. B62 and A.f. 3357 was calculated as previously published [37]:PI = [(C − T)/C] × 100
where PI is the percent inhibition; C is the measurement of the colony radius of the aflatoxigenic strain on the control; and T is the measurement of the colony radius of the aflatoxigenic strain in the co-culture.

The type of fungal interaction was examined after 5 days of point inoculation and described using a previously published rating scale by Whipps et al. (1987) [44]. Type 0 for no observed interaction; Type 1 when *Trichoderma* mycelium overgrows the *Aspergillus* colony and *Aspergillus* growth stops; Type 1/2 when *Trichoderma* overgrows *Aspergillus* colony, but *Aspergillus* continues to grow; Type 2/1 when *Aspergillus* mycelium overgrows *Trichoderma* colony but *Trichoderma* continues to grow; Type 2 when *Aspergillus* mycelium overgrows *Trichoderma* colony and *Trichoderma* growth stops; Type 3 when a slight mutual inhibition was observed, as evidenced by the presence of an inhibition zone ≤ 2 mm distance between colonies; and Type 4 when extreme inhibition was observed by the presence of an inhibition zone ≤ 4 mm distance between colonies. A type “0” was added to the original scale by Whipps et al. (1987) to describe the absence of interaction or contact between *Trichoderma* and *Aspergillus*.

### 5.3. Sandwiched Plate Assay for VOC-Mediated Interaction

Sandwiched Petri plates were used to determine whether *Trichoderma* VOCs affect the growth of A.p. B62 and A.f. 3357, as previously published [28]. The center of a PDA plate was center inoculated with either *Trichoderma* or *Aspergillus* with 5 μL of 10^6^ spores/mL (Figure 5B). Plates were left to equilibrate for 15 min at room temperature. The plate with *Aspergillus* was placed on top of a *Trichoderma* plate, sealed with Bemis™ Parafilm™ M (Fisher Scientific, Hampton, NH, USA) and placed into a Quart-size Ziplock bag to minimize VOC transfer between treatments. Sandwiched Petri plates were incubated at 30 °C for 5 days. For control, each *Trichoderma* species, A.p. B62, and A.f. 3357 were sandwiched with 5 μL of sterile DI water center inoculated onto PDA. The fungal colony diameter of the control and sandwich treatment was measured daily for 5 days. Triplicate plates were set up for each control/sandwich treatment with three independent biological experiments (total, *n* = 9).

### 5.4. Cellophane Assay for Non-Volatile Compounds Mediated Interaction

A sterilized cellophane membrane (Idea Scientific Company, Minneapolis, MN, USA) was fitted on top of a PDA plate, as previously published [28]. Plates with cellophane were left overnight to allow excess water to evaporate at room temperature. Five μL of *Trichoderma* (1 × 10^6^ spores/mL) was center inoculated onto PDA overlaid with a cellophane membrane. Five μL of sterile DI water was used for control plates (Figure 5C). Cellophane and control plates were incubated at 30 °C for 48 h to allow the secreted metabolites (non-VOC) from *Trichoderma* to be released into PDA through the cellophane, with the molecular weight cut off of 3.5 kDa [28]. Cellophane containing *Trichoderma* or water control was discarded, followed by center inoculation with 5 μL of A.f.3357 or A.p. B62 (final spore concentration = 5 × 10^3^ spores). Cellophane and control plates were incubated at 30 °C for 5 days. *Aspergillus* colony diameter of control and cellophane treatment was measured daily for 5 days. Triplicate plates were set up for each control/cellophane treatment with three independent biological experiments (total, *n* = 9).

### 5.5. Examination of AF Levels in A. parasiticus and A. flavus by TLC

Aflatoxin production was assessed using thin layer chromatography (TLC) analysis, carried out as published [48]. Briefly, AF produced from PDA containing 5-day fungal colonies for the control and treatment of the sandwich and cellophane assays were extracted three times with 5 mL chloroform (total, 15 mL chloroform). Chloroform was evaporated and resuspended in 1 mL of 70% methanol. Equal amounts (10 μL) of methanolic extracts were spotted onto a 20 × 20 cm Silica gel 60F254 glass TLC plate (EMD Millipore Corporation, Burlington, NJ, USA). The aflatoxin standards used were as follows: AFB1 (Enzo Life Sciences, Farmingdale, NY, USA), AFB2 (Cayman Chemical Company, Ann Arbor, MI, USA), and AFG1 (Sigma-Aldrich, St. Louis, MO, USA). All standards were dissolved in 70% methanol with a final concentration of 2 mg/mL and mixed in equal volumes (1:1:1). A 10 μL solution of AF mixture was used as a positive control. The spots were allowed to dry, and the TLC plate was developed in chloroform–acetone (9:1, *v*/*v*). Aflatoxin levels were visualized using the UV light setting on a Molecular Imager^®^ Gel Doc™ XR System (Bio-Rad Laboratories, Hercules, CA, USA).

### 5.6. Statistical Analysis

The statistical analyses were performed using Microsoft Office Excel 2010 (Microsoft Co., Redmond, WA, USA) and GraphPad Prism version 10.1.0 (GraphPad Software, Boston, MA, USA). Data for all three models were presented as bar graphs representing mean ± SEM of colony growth and % growth inhibition. One-way analysis of variation (ANOVA) was performed using GraphPad Prism version 10.1.0. The significance of each treatment was determined using the F value. When a significant F test was observed, the separation of the means was carried out using Dunnett’s and Tukey HSD’s tests. Statistical significance was determined at *p* ≤ 0.05.

## Figures and Tables

**Figure 1 toxins-16-00314-f001:**
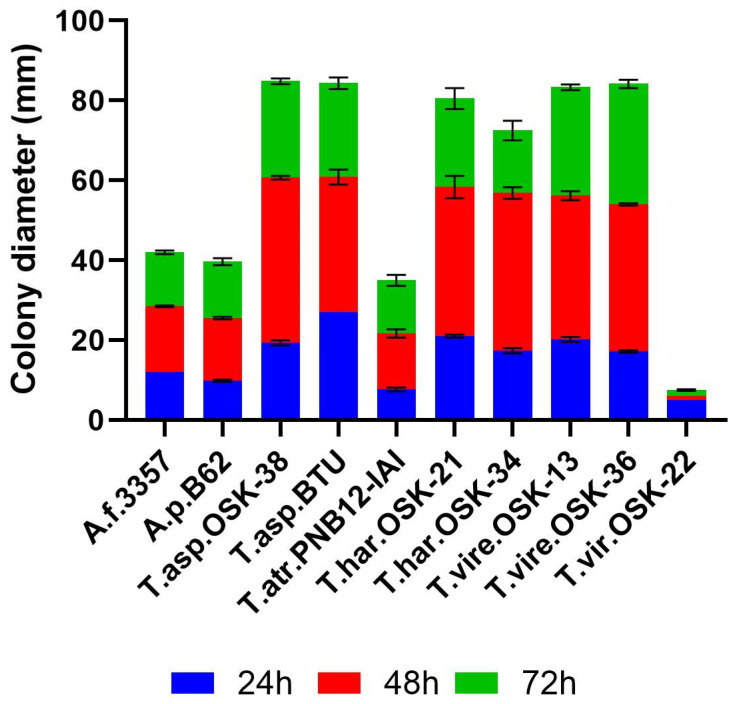
Growth rate as a prerequisite of strain fitness. Bar graphs represent mean colony diameter (in mm) measured at 24, 48, and 72 h for all strains (Appendix A) grown on PDA at 30 °C. Blue represents 24 h, red represents 48 h, and green represents 72 h. Spores were center inoculated onto PDA with a final concentration of 5 × 10^3^ spores. Error bars represent mean colony diameter ± SEM (*n* = 6).

**Figure 2 toxins-16-00314-f002:**
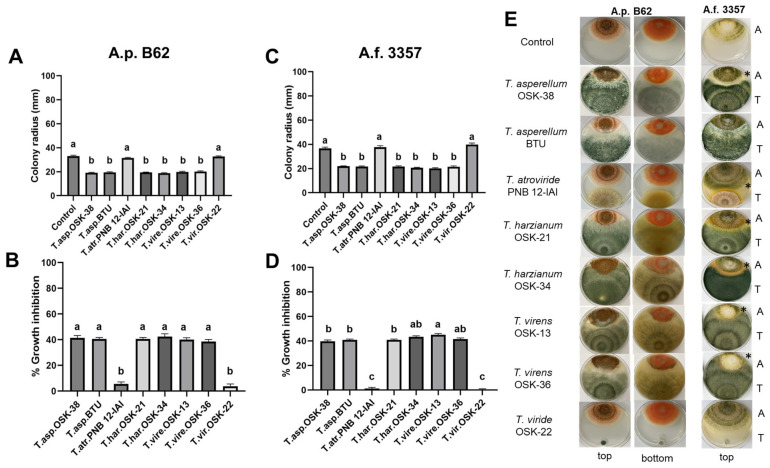
Fungal growth as a function of colony radius (**A**,**C**) and percent inhibition (**B**,**D**) of *Trichoderma* spp. with A.p. B62 (left panel) and A.f. 3357 (right panel) on dual culture plate interaction assays at 120 h on PDA at 30 °C. Bar graphs represent the mean colony radius or % growth inhibition ± SEM (*n* = 9). Different letters indicate significant differences between the strains from the control (**A**,**C**) and from each other (**B**,**D**) based on Dunnett’s and Tukey HSD’s test, respectively (*p* < 0.05). (**E**) Representative images of fungal colonies of A.p. B62 and A.f. 3357 without the presence of *Trichoderma* (Control) and with all eight *Trichoderma* spp. Top and bottom view of fungal colonies were imaged for A.p. B62 due to the presence of norsolorinic acid, an orange pigment that is used as an indicator of aflatoxin; top view only is presented for A.f. 3357. Type 3 and 4 interactions are marked with an asterisk (*) indicating the presence of a zone of inhibition. The letter ‘A’ represents the location where aflatoxigenic strains were point inoculated, while the letter ‘T’ indicates where *Trichoderma* was inoculated.

**Figure 3 toxins-16-00314-f003:**
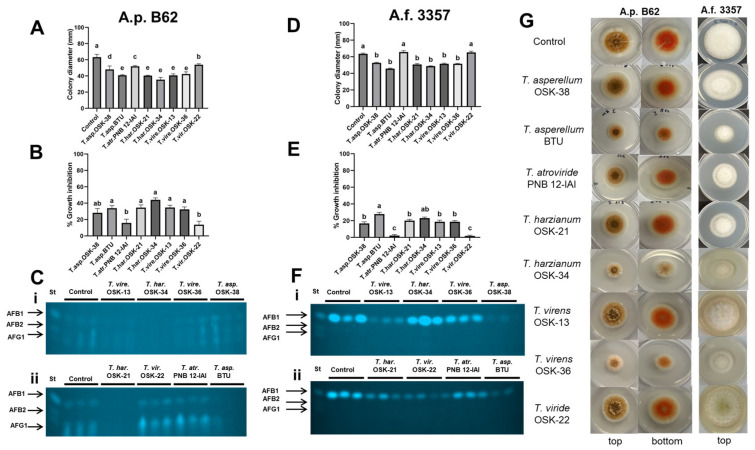
Sandwiched plate assay comparing the effect of *Trichoderma* VOCs on *Aspergillus* growth as measured by (**A**,**D**) colony diameter and (**B**,**E**) % growth inhibition, and (**C**,**F**) AF levels by TLC. Bar graphs represent mean colony diameter or % growth inhibition ± SEM (*n* = 9). Different letters indicate significant differences between the strains from the control (**A**,**D**) and from each other (**B**,**E**) based on Dunnett’s and Tukey HSD’s test, respectively (*p* < 0.05). Aflatoxins were extracted from the sandwiched plate assay of A.p. B62 and A.f. 3357 control and treatment at 120 h and visualized by TLC (**C**,**F**). ‘i’ indicates top TLC image, ‘ii’ indicates bottom TLC image. Lane St contains 20 µg of AFB1, AFB2, and AFG1 standard mixture. (**G**) Representative images of fungal colonies of A.p. B62 and A.f. 3357 without the presence of *Trichoderma* (Control) and with all eight *Trichoderma* spp. The top and bottom views of fungal colonies were imaged for A.p. B62 due to the presence of norsolorinic acid, an orange pigment that is used as an indicator of AF; top view only is presented for A.f. 3357.

**Figure 4 toxins-16-00314-f004:**
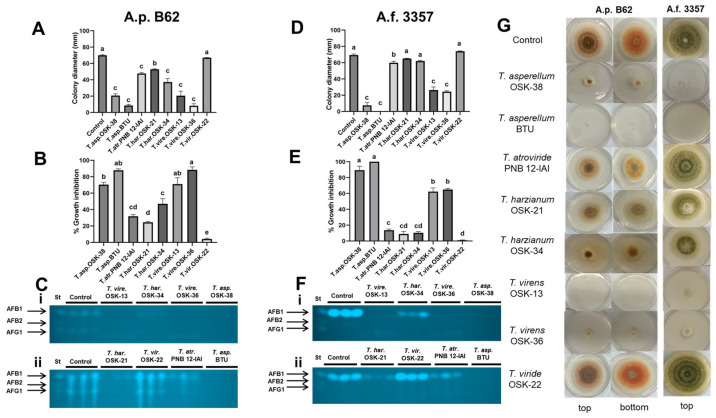
Cellophane membrane-based interaction assay comparing the secreted metabolites of *Trichoderma* at 48 h on (**A**,**D**) colony diameter, (**B**,**E**) growth inhibition, and (**C**,**F**) AF levels by TLC. Bar graphs represent mean colony diameter or % growth inhibition ± SEM (*n* = 9). Different letters indicate significant differences between strains from the control (**A**,**D**) and from each other (**B**,**E**) based on Dunnett’s and Tukey HSD’s tests, respectively (*p* < 0.05). Aflatoxins were extracted from the cellophane membrane-based assay of A.p. B62 and A.f. 3357 control and treatment at 120 h and visualized by TLC (**C**,**F**). ‘i’ indicates top TLC image, ‘ii’ indicates bottom TLC image. Lane St contains 20 µg of AFB1, AFB2, and AFG1 standard mixture. (**G**) Representative images of fungal colonies of A.p. B62 and A.f. 3357 control and with all eight *Trichoderma* spp. The top and bottom views of fungal colonies were imaged for A.p. B62 due to the presence of norsolorinic acid, an orange pigment that is used as an indicator of AF; top view only is presented for A.f. 3357.

**Figure 5 toxins-16-00314-f005:**
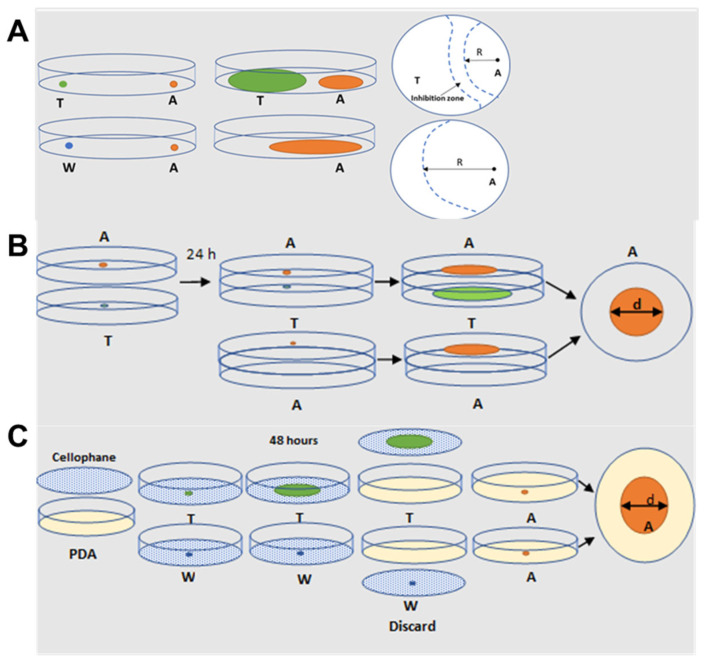
Three ecological models for studying the interaction between *Trichoderma* spp. with aflatoxigenic A.f. 3357 and A.p. B62. A schematic representation of the effect of (**A**) physical interaction between *Trichoderma* and *Aspergillus* using a dual culture plate interaction assay, (**B**) *Trichoderma* volatiles on *Aspergillus* using a sandwich plate assay, and (**C**) *Trichoderma* non-volatile compounds or secreted metabolites on *Aspergillus*. Abbreviations: T—*Trichoderma* strain, A—*Aspergillus* strain, W—sterile DI water control, R—radius, d—diameter; orange circles—A.p. B62 growth, green circles—*Trichoderma* growth.

**Table 1 toxins-16-00314-t001:** Growth inhibition and classification of the interaction type between *Trichoderma* and *Aspergillus* in the dual culture plate assay on PDA at 30 °C for 5 days.

*Trichoderma*	A.p. B62		A.f. 3357	
	% I ^a^	Interaction Type ^b^	% I ^a^	Interaction Type ^b^
*T. asperellum* OSK-38	41.5 ± 1.8	1	39.8 ± 1.1	4
*T. asperellum* BTU	40.7 ± 1.1	1	40.9 ± 0.8	1/2
*T. atroviride* PNB 12-IAI	5.5 ± 1.6	0	1.2 ± 0.9	3
*T. harzianum* OSK-21	40.6 ± 1.2	1	40.9 ± 0.8	3
*T. harzianum* OSK-34	42.4 ± 2.2	1/2	43.5 ± 0.8	4
*T. virens* OSK-13	40.1 ± 1.5	1/2	45.1 ± 0.9	3
*T. virens* OSK-36	38.6 ±1.5	1/2	41.6 ± 0.9	3
*T. viride* OSK-22	3.8 ± 1.7	0	0.6 ± 0.4	0

^a^ Percent growth inhibition of *Aspergillus* in the presence of *Trichoderma* presented as mean ± SEM (*n* = 9). ^b^ Interaction type as described by Whipps et al. (1987) [44] (see Section 5).

## Data Availability

The original contributions presented in the study are included in the article/Appendix A, further inquiries can be directed to the corresponding authors.

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
