# Peer review of "Three Ecological Models to Evaluate the Effectiveness of Trichoderma spp. for Suppressing Aflatoxigenic Aspergillus flavus and Aspergillus parasiticus"

_toxins, 2024, doi:10.3390/toxins16070314_

Round 1

Reviewer 1 Report

Comments and Suggestions for Authors

Dear Author,

Manuscript is well written and clearly describe the significant finding with suitable experimental proof in the form of table and figures. The manuscript insight on interactions of Trichoderma with mycotoxins producing fungal strains in different ways. 

Please see the comments in attached file.

I suggest including to the figures for the 1. Dual culture plate interaction assay, 2. Sandwich assay, and 3. Cellophane assay in the main manuscript instead of the supplementary material. Presenting them as a single figure in the main text will enhance the readers' understanding of your experiments.

Reviewer 2 Report

Comments and Suggestions for Authors

Question1(Q1)

Q2.   Line 9 "Control" can be replaced with "management" to avoid redundancy.

Q3.   L15 “antibiosis" and "mycoparasitic behavior” are two words that need to clarification for resers who might not familiar with these terms.

Q4.   L16 "volatile organic compounds" could be introduced the abbreviation “VOCs”, which is used later.

Q5.   L6 “continue” should be “continues” to maintain subject-verb agreement

Q6.   L7 “occurrence” should be “occurrences” for consistency in plural form

Q7.   "dual plate interaction assay suggest" should be "dual plate interaction assay suggests" for subject-verb agreement.

Q8.   "The economic impact of mycotoxin is" should be "The economic impact of mycotoxins is" for correct subject-verb agreement and to reflect plural form.

Q9.   "contamination on food and feed crops are heavily regulated" should be "contamination in food and feed crops is heavily regulated" for subject-verb agreement.

Q10. "Aflatoxins (AF) are a group of naturally occurring secondary metabolites" would be clearer as "Aflatoxins (AFs) are a group of naturally occurring secondary metabolites" to introduce the abbreviation correctly. Check the whole article to correct the mistakes.

Q11. The phrase "1 billion metric tons of food losses are attributed to mycotoxin contamination globally" can be simplified to "1 billion metric tons of food are lost globally due to mycotoxin contamination."

Q12. L69 "strain" should be "strains" for subject-verb agreement.

Q13. L75 The phrase "competitive exclusion of toxigenic strains" is redundant and can be simplified.

Q14. L85 Missing period after “polyketides”

Q15. L104 “competitive exclusion …” should be “competitive exclusion as a results of the utilization of limited resources”

Q16. L 82-83: "volatile or non-volatile organic compounds with antifungal, antibiotic, and growth inhibition properties".Could be simplified to "organic compounds with antifungal and antibiotic properties.

Q17. Line 75-77 , Line 110-112 The sentence is overly complex and could be broken down for clarity.

Q18. L 159: "highest % growth inhibition" Should be "highest percentage growth inhibition."

Q19. L 164: "Trichoderma species mutually developed together" Redundant; use either "mutually developed" or "developed together."

Q20. L140 The transition between T. viride OSK-22's growth rate and the growth rate of fast-growing strains could be clearer.

Q21. L135 "colony diameter over 24, 48, and 72 hours under standard laboratory conditions"

should specify the conditions more precisely if they differ in any way.

Q22.  L296 The transition between discussing antibiosis and mycoparasitism could be smoother.

Q23. L325 The statement lacks specificity; it could be clearer about the purpose of comparing VOC profiles.

Q24. L300 "Trichoderma mycelium growing over the aflatoxigenic strains."Growing" should agree with the plural "strains"; it should be "Trichoderma mycelium grows over the aflatoxigenic strains."

Q25. L307 "AF levels in B62 is lower than in A.f. 3357.""Is" should be "are" to match the plural subject "AF levels

Q26. L313 VOC" should be pluralized as "VOCs."

Q27. The conclusion could be more specific about the key insights or findings derived from the three models discussed.

Q28. L307 The statement could be clearer about how these models aid  in the screening process and what benefits they offer over field studies.

Comments on the Quality of English Language

could be improved

Reviewer 3 Report

Comments and Suggestions for Authors

The manuscript is well-written, the methodology is correct and the results look realistic. There is an appropriate discussion on the weak points of the study, that is, the translation of the models beyond the in vitro level, and the TLC analysis. I have minor comments.

1) Is it possible to try at least a spectrophotometric analysis instead of TLC for some samples? In this case, you may have a calibration series and can measure the extracted AF (at the wavelength corresponding to the absorption maximum of AF) using an internal standard (standard addition) procedure. It may work if the matrix effect is not very large. Of course, HPLC analyses would be the best...

2) Maybe, it would be useful to check the direct effect of some of the VOCs  (or non-volatiles) that are suspected to participate in the anti-fungal effect. It may be done by the application of solutions of different VOCs (or non-volatiles) to see which compound is the most active in this particular case. (Or maybe there is data on this question in the literature?)

Reviewer 4 Report

Comments and Suggestions for Authors

Manuscript Three ecological models for the study of the interaction between Trichoderma spp. with aflatoxigenic Aspergillus flavus and Aspergillus parasiticus provides data on 8 strains of Trichoderma as inhibitors of toxigenic A. flavus and A. parasiticus obtained by three models of competitive exclusion. In my opinion the manuscript lacks novelty in the findings of the competitive exclusion mechanism, similar experiments have been carried out in the cited references, e.g. 35, 39, 47, 48. As the authors noted, GC/MS analysis of volatile compounds or LC/MS of agar eluted compounds would bring news to this topic. Also, designing an experiment that can simulate field conditions would bring novelty to the topic. In my opinion, the manuscript is worthy of publication as a technical paper because it provides additional data on the antagonism of Trichoderma and toxigenic Aspergillus / microparasitic relationships.

Here are some specific comments:

- The summary is too long, it has 260 words

- The introduction should be shorter: we can exclude lines 35-39 because it is too general and not relevant to the manuscript; lines 63-77 describe a competitive exclusion strategy using colonization with non-aflatoxigenic A. flavus strains to exclude toxigenic A. flavus strains on different crop samples, these lines can be omitted as they are not relevant to the work; lines 119-128 should be in the Results, not the Introduction

Round 2

Reviewer 2 Report

Comments and Suggestions for Authors

The manuscript has been significantly improved by the authors. I do not have other comments.

Comments on the Quality of English Language

Some minor errors should be corrected.

Author Response

[Reviewer comments]: The manuscript has been significantly improved by the authors. I do not have other comments.

Some minor errors should be corrected.

[Author response]: We have reviewed the manuscript and corrected minor errors. We thank the reviewer for their time and support.

Reviewer 4 Report

Comments and Suggestions for Authors

The authors have responded to all comments, and some responses I do not consider acceptable. I still think that the general sentences about the contamination of food by moulds that produce mycotoxins and cause economic losses can be omitted because such sentences have been written in thousands of articles dealing with the issue of toxigenic molds. Those sentences neither contribute to the quality nor the interestingness of the manuscript.

The abstract should be shortened to 200 words, as it says in the Instructions for authors.

I advise authors to read Toxins journal instructions on writing an introduction: highlight controversial and diverging hypotheses when necessary. Finally, briefly mention the main aim of the work and highlight the main conclusions. In accordance with the instructions, I advise the authors to remove the results from the introduction and write main conclusion after the hypothesis and the methodology they used to prove the hypothesis. In the book that the authors cited in their responses to the reviewer (Chapter 10 of Robert A. Day and Barbara Gastel’s book on How to write and publish a scientific paper), there is also an instruction for a good introduction: It should briefly review the pertinent literature to orient the reader. It should also identify the gap in the literature that the current research was intended to address. It should then make clear the objective of the research. 

The changes made by the authors improved the manuscript. In my opinion, the manuscript can be published after minor changes, as I stated in the previous text

Author Response

Round #2

[Reviewer comment 1]: The authors have responded to all comments, and some responses I do not consider acceptable. I still think that the general sentences about the contamination of food by moulds that produce mycotoxins and cause economic losses can be omitted because such sentences have been written in thousands of articles dealing with the issue of toxigenic molds. Those sentences neither contribute to the quality nor the interestingness of the manuscript.

[Author response 1]: We understand and agree with the reviewer, thank you for comment. We have removed and simplified the introduction and focused on aflatoxin and aflatoxigenic Aspergillus spp. This removed content that is not relevant and corresponding references. This makes the first paragraph in the introduction much more focused. See revised manuscript L30-45.

[Reviewer comment 2]: The abstract should be shortened to 200 words, as it says in the Instructions for authors.

[Author response 2]: Abstract has been reduced to 200 words. We apologize for this oversight. See revised manuscript, L6-21.

[Reviewer comment 3]: I advise authors to read Toxins journal instructions on writing an introduction: highlight controversial and diverging hypotheses when necessary. Finally, briefly mention the main aim of the work and highlight the main conclusions. In accordance with the instructions, I advise the authors to remove the results from the introduction and write main conclusion after the hypothesis and the methodology they used to prove the hypothesis. In the book that the authors cited in their responses to the reviewer (Chapter 10 of Robert A. Day and Barbara Gastel’s book on How to write and publish a scientific paper), there is also an instruction for a good introduction: It should briefly review the pertinent literature to orient the reader. It should also identify the gap in the literature that the current research was intended to address. It should then make clear the objective of the research.

[Author response 3]: Thank you for this comment. We agree with the reviewer and removed results from the introduction, wrote the main conclusion after the hypothesis and the methodology we used to test the hypothesis. The last paragraph of the introduction now reads much clearer:

The aim of this work is to characterize Trichoderma-Aspergillus interactions through physical contact, volatile-mediated interaction, and through secreted metabolites. We hypothesized that Trichoderma with the highest growth rates would be most successful across all three models. We further hypothesized that there would be strain-specific differences in the effect of Trichoderma VOC and non-VOC on Aspergillus growth and toxin production. Thus, we tested three ecological models to evaluate the effectiveness of Trichoderma spp. for suppressing aflatoxigenic A. flavus and A. parasiticus. Fungal growth and aflatoxin levels were used to evaluate the effectiveness of the dual plate assay (physical interaction), sandwiched plate assay (VOC-mediated interaction), and the cellophane membrane-based assay (interaction via secreted metabolites). Overall, T. asperellum BTU was identified as most effective in suppressing the growth and aflatoxin B1 of A. flavus and A. parasiticus across all models.

See revised manuscript, L93-104.

[Reviewer comment #4]: The changes made by the authors improved the manuscript. In my opinion, the manuscript can be published after minor changes, as I stated in the previous text

[Author response #4]: We have addressed all additional comments by the reviewer and thank the reviewer for their time and support.
